# Parasite Prevalence May Drive the Biotic Impoverishment of New England (USA) Bumble Bee Communities

**DOI:** 10.3390/insects12100941

**Published:** 2021-10-16

**Authors:** Anne L. Averill, Andrea V. Couto, Jeremy C. Andersen, Joseph S. Elkinton

**Affiliations:** 1Department of Environmental Conservation, University of Massachusetts Amherst, Amherst, MA 01003, USA; jcandersen@umass.edu (J.C.A.); elkinton@umass.edu (J.S.E.); 2Department of Computer Science, Bridgewater State University, Bridgewater, MA 02324, USA; andrea.v.couto14@gmail.com

**Keywords:** *Bombus*, *Crithidia*, *Nosema*, pathogens, pollinator health, community ecology, competition

## Abstract

**Simple Summary:**

Here we discuss widespread changes in the community structure of bumble bees (*Bombus* spp.) found in the coastal-zone community of New England. One species in particular, *Bombus impatiens* Cresson, 1863, has increased in relative abundance nearly 45% since the 1990s to become the dominant species in the region, representing nearly 75% of all *Bombus* individuals collected in our studies. These changes in abundance may be, in part, due to differences in infection rates by microparasites, with *B. impatiens* having significantly fewer microparasites than several other less common and declining *Bombus* species. We discuss the possible role of microparasites in influencing the community composition of *Bombus* species in our region, and how these infections might be compounding declines in conjunction with habitat loss and climate change.

**Abstract:**

Numerous studies have reported a diversity of stressors that may explain continental-scale declines in populations of native pollinators, particularly those in the genus *Bombus*. However, there has been little focus on the identification of the local-scale dynamics that may structure currently impoverished *Bombus* communities. For example, the historically diverse coastal-zone communities of New England (USA) now comprise only a few species and are primarily dominated by a single species, *B.* *impatiens*. To better understand the local-scale factors that might be influencing this change in community structure, we examined differences in the presence of parasites in different species of *Bombus* collected in coastal-zone communities. Our results indicate that *Bombus* species that are in decline in this region were more likely to harbor parasites than are *B. impatiens* populations, which were more likely to be parasite-free and to harbor fewer intense infections or co-infections. The contrasting parasite burden between co-occurring winners and losers in this community may impact the endgame of asymmetric contests among species competing for dwindling resources. We suggest that under changing climate and landscape conditions, increasing domination of communities by healthy, synanthropic *Bombus* species (such as *B. impatiens*) may be another factor hastening the further erosion of bumble bee diversity.

## 1. Introduction

Worldwide decline of bumble bee (*Bombus* spp. Latreille, 1802 [Hymenoptera: Apidae]) populations is ongoing reviewed in [1], and there is accumulating evidence that *Bombus* species may be particularly vulnerable to anthropogenic disturbances [2]. These shifts in diversity may pose a challenge to the functioning of many natural and agricultural ecosystems, where bumble bees are the predominant wild pollinators [3,4,5]. For example, wild bumble bees are by far the most abundant and effective pollinators of the New England (USA) cranberry crop, *Vaccinium macrocarpon* Aiton (Ericales: Ericaceae) [6,7,8], but the impoverishment of *Bombus* diversity in the Massachusetts growing region has been profound over the past half century. Henry J. Franklin, who completed a comprehensive revision of North American bumble bees published in 1913 [9], regularly observed nine species in the southeastern Massachusetts cranberry landscape [10]. However, five of these species are now rare or extirpated [8], and further declines in species evenness is ongoing. Our long-term surveys in southeastern Massachusetts began in 1990 [11] and used a repeated collection procedure at the same sites. We found that species loss is matched by the rapid increase in *B. impatiens* Cresson, 1863, with worker captures of *B. impatiens* now dominating all collections [8,12]. In our long-term survey, we also quantified bumble bee visits per minute at flowering cranberry beds in late spring-early summer and found that overall abundance has not declined [12]. It thus seems reasonable to conclude that *B. impatiens*, has backfilled vacated niche space of the lost species, and represents not only a greater relative abundance but also a greater absolute abundance in our surveyed communities.

Increases in the relative abundance of *B. impatiens* are not unique to the cranberry agroecosystem and have been documented across the northeastern United States [2,13,14]. This may be due to competition for forage and nest resources or emigration of imported cultures of *B. impatiens* into the surrounding natural landscapes. However, in a previous study in this system, Suni et al. [15] found no evidence of widespread introgression of alleles from commercial *B. impatiens* to wild bees and no evidence that commercial bumble bees were becoming established. There is increasing evidence that mass-reared commercial *B. impatiens* colonies imported to farms and greenhouses for pollination in the Pacific Northwest (USA) are responsible for the widespread observation of range expansion where *B. impatiens* is non-native [16]. In other regions, introduction of commercial European *Bombus* species, particularly *Bombus terrestris* L., 1758 for pollination services may be responsible for contraction and displacement of native species [17,18]. While the mechanism by which *B. impatiens* is replacing other species in our study region remains unknown, it is an adaptable, synanthropic species [19] that maintains large populations in many landscapes [20]. The species has a broad floral breadth, increased pesticide tolerance, and an early emergence coupled with a long flight season [5]. Our region has experienced warmer and wetter conditions than recorded historically, and that may accelerate extinction of some *Bombus* species [20]. On the other hand, large and widespread colonies of *B. impatiens* may be buffered from loss of genetic diversity and demographic stochasticity [21] and thus be resilient to the challenges posed by climate change. Finally, another factor that might be promoting the increased abundance of *B. impatiens* in this region is parasite load. *Bombus* individuals may be infected with a range of parasites (here the term parasite is used to include pathogens, parasites, and parasitoids) that have negative impacts on aspects of *Bombus* biology [22]. It has become a rule of thumb that higher prevalence of the microsporidia *Nosema bombi* Fanthan & Porter is an indicator of species decline reviewed in [23]. However, the cause for varying *Nosema* levels among species is open to different interpretations. Species may naturally vary in susceptibility to the pathogen or alternatively, higher levels may be seen in isolated and inbred populations in decline [24]. However, in some European surveys, widespread and common species, such as *B. terrestris* and *B. pascuorum* (Scopoli), 1763, often are found with high microparasite infection reviewed in [25]. In addition to *N. bombi*, the trypanosomatid *Crithidia bombi* Lipa & Triggiani has also been observed infecting species of *Bombus* [26,27,28]. Interestingly, this parasite is often associated with common *Bombus* species [23,29], including *B. impatiens*, where in one study infection levels reached over 60% [29].

We initiated an intensive regional investigation of comparative prevalence of parasites in increasing vs. declining *Bombus* species. In this study, we quantified parasites seen in *Bombus* community members in the coastal-zone region of Massachusetts across the entire season of bumble bee activity. Members of this community included *B. impatiens,* as well as several species of *Bombus* that vary in conservation status [12, summarized in Table 1) including; *B. vagans* Smith, 1854 (in strong decline), *B. griseocollis* (DeGeer), 1773 (gradually increasing its distribution and abundance), *B. bimaculatus* Cresson, 1863 (apparently stable in distribution, but with variable abundance among years), and *B. perplexus* Cresson, 1863 (at the onset of decline). Based on our observations obtained over several years that *B. impatiens* was increasingly dominating survey samples and dissections seldom revealed intestinal pathogens compared to rarer species, we hypothesized that overall colony health of *B. impatiens* may also be a factor favoring rapid increase in populations.

## 2. Materials and Methods

### 2.1. Study Area

We established 73 collection sites in southeastern Massachusetts, which encompassed forested, urban and suburban areas as well as ca. 5000 ha of cultivated cranberry (Figure 1). Only two sites were at cranberry farms, and there was no other significant agriculture in the region. The ca. 50 km^2^ sampled area was in two sub-ecoregions, Bristol Lowlands and Cape Cod and the Islands, within the Northeastern coastal zone. This zone has nearly no flower-rich grasslands or maintained meadows (favored by many *Bombus* spp. [1]) and is characterized by sandy and acidic soils that support *Pinus rigida* Mill. and *Quercus ilicifolia* Wangenh forests [30]. About half of the sites were located within the region’s pine barrens, and as a result, the majority of flowering plants where bees were collected occurred near buildings, gardens, roadsides, and other public rights of way.

### 2.2. Collections

From 18 May to 4 November 2015, 2–3 trained collectors visited sites between the hours of 0900 and 1600 on days with no precipitation and with temperatures ranging from 22–30 °C. Sampling was conducted at a site one to eight times over the season, and on average, four times. All but six of the sites were visited multiple times. For 10 min each, the collectors quickly moved among flower patches up to 100 m from the starting point to capture as many bees as possible in plastic vials. Bees were identified in the lab using keys in Williams et al., 2014 [19].

### 2.3. Cryptic Bombus Identifications

Recent work has shown that field identifications for three cryptic species of *Bombus* found in eastern North America (*Bombus sandersoni* Franklin, 1913, *B. vagans*, and *B. perplexus*) may not be possible without calculations of malar ratios or DNA barcoding [31]. To verify that our identifications for individuals identified as *B. vagans* were accurate, we obtained species-level identifications based on the amplification of a fragment of the mitochondrial locus cytochrome oxidase I (COI) for a subset of identified *B. vagans* individuals following the methods presented in [31]. Forward and reverse sequences were produced at the DNA Analysis Facility on Science Hill at Yale University, and the results edited in Geneious V. 11.1.2 (Biomatters Ltd., Auckland, New Zealand). Consensus sequences were compared to published sequences in the GenBank database using the ‘blastn’ algorithm implemented through https://blast.ncbi.nlm.nih.gov/ (accessed on 17 March 2021).

### 2.4. Parasites and Their Assessment

We quantified the presence of three intestinal microparasites based on visual examinations: microsporidia *(Nosema* sp.), neogregarines, and trypanosomes, as well as unidentified species of nematodes (Mermithidae), and an unidentified endoparasitic conopid fly (Diptera: Conopidae). In studies on susceptible European species, *Nosema* is considered a high impact parasite [22,32,33,34]. In contrast, trypanosomes such as *C. bombi* causes many sublethal effects [35,36,37,38,39,40,41]. Both *C. bombi* and *N*. *bombi* have been determined to be transmitted horizontally via an oral-fecal route, among flowers and within the colony [42,43,44], and while it is currently unknown, this is probably also true for neogregarines such as *A. bombi* [34]. The conopid parasitoid larvae were not reared out or identified, but previous studies conducted in this region identified all conopids as members of the genus *Physocephala* [29,45]. Similarly, to the results presented in Kissinger et al. [27], nematodes were only found at very low numbers; as such, they were not identified.

To determine whether or not collected bees were infected, the presence of the different parasites described above was assessed through gut dissections. Prior to dissection, live bees were placed in a freezer until immobilized. The gut was then completely removed and inspected for conopid larvae and nematodes. To assess the intestinal microparasites, a mixture of gut fluids plus gut tissues and malpighian tubules were minced in 50 μL of water on a microscope slide. Dissection tools were flame-sterilized between each sample, and the presence or absence of each parasite was quantified by randomly inspecting five visual fields on each slide under phase-contrast microscopy (400× magnification). Because we chose to dissect the bees while alive on the day captured, to maximize efficiency, we determined high-intensity infections in a non-exacting manner by recording when one or more visual fields were flooded with cells or spores of a pathogen.

Prevalence, or the proportion of infected bees among all the bees examined, were compared by Fisher’s Exact Test. Since male captures were low for all species except *B. impatiens* and *B. bimaculatus*, only workers were included for comparisons of parasite prevalence among species. A separate comparison was made between *B. impatiens* and *B. bimaculatus* males and workers. Because neogregarine and nematode infections were found infrequently, individuals with these two infections were pooled and comparisons were made between groups of bee species.

## 3. Results

### 3.1. Collections

A total of 1205 bumble bees were collected. All species overlapped in captures during a 1.5-month (June–July) period. In total, bees were collected from 58 flowering plants, with all five *Bombus* species collected while foraging on *Rhododendron* spp. and *Nepeta xfaasenii*, and four of the five species collected while foraging on *Spirea japonica, Centaurea maculosa,* and *Salvia nemerosa.* Field identifications found that the most abundant species was *B. impatiens* (74.9% of the collection), followed by *B. bimaculatus* (9.9%:), *B. perplexus* (8.2%), *B. vagans* (5.1%), and *B. grisceocollis* (1.7%). Three *B. fervidus* (Fabricius) workers were also collected, historically a common species in the region [9,46], but currently extremely rare [8]. Comparisons of these identifications to those obtained through DNA barcoding of 61 field identified *B. vagans* individuals resulted in the generation of clean sequence reads from 28 of the samples. All of the *B. vagans* identifications were confirmed with DNA barcoding (overall accuracy of identifications for this group was thus 100%). DNA sequences generated in this study are available on GenBank (accession numbers OK044437-65).

### 3.2. Parasites and Their Assessments

*Bombus impatiens* individuals were more likely to be free of all parasites (69.2% were parasite-free) when compared to *B. bimaculatus, B. perplexus,* and *B. vagans*, a grouping that was 43.4% parasite-free (*p* < 0.0001; FET). Moreover, of the 54 bees with multiple parasites per individual, *B. impatiens* was least likely to harbor two or more parasite species (3.4%) when compared to the prevalence of co-infections of *B. bimaculatus* (9.4%, *p* = 0.0083; FET), *B. perplexus* (11.1%, *p* = 0.0018; FET), and *B. vagans* (15.3%, *p* = 0.0004; FET). Looking at the three microparasites and across the collection, intense infections were recorded for 67 bees and the combined group of *B. bimaculatus, B. perplexus,* and *B. vagans* (15.1%) was significantly more likely to have at least one intense infection compared to *B. impatiens* (2.7%, *p* < 0.0001; FET). *Bombus griseocollis* had no intense infections (Table 1).

For the three microparasite groups combined, 86.6% of *B. impatiens* workers and 89.5% of *B. griseocollis* were uninfected, while about half as many *B. bimaculatus* (45.8%), *B. perplexus* (43.3%), and *B. vagans* (47.6%) were free of these pathogens (Table 1). For individual pathogen prevalence (Table 2; Figure 2), *B. impatiens* individuals had significantly fewer trypanosomatid infections (11.0%) than *B. vagans* (32.2%), *B. bimaculatus* (48.5%) and *B. perplexus* (49.5%) (for all comparisons with *B. impatiens*, *p* < 0.0001; FET). *Nosema* infection levels for *B. impatiens, B. bimaculatus* and *B. griseocollis* were similar. While *Nosema* prevalence in *B. impatiens* (4.7%) was lower than in *B. perplexus* (11.1%), this difference was marginally significant (*p* = 0.015; FET), in contrast to a comparison of *Nosema* infection of *B. vagans* (20.3%), where the difference with *B. impatiens* was highly significant (*p* < 0.0001; FET). No *Nosema* spores were observed in *B. bimaculatus*, even though a large sample of workers (107) was evaluated. For the three *B. fervidus* individuals collected, each was from a different site; two individuals were infected with *Nosema*, but with no other parasite.

Neogregarines were detected in 30 workers (3.0%) across the entire collection. Overall prevalence of neogregarines were significantly higher in *B. bimaculatus* (9.35%) when compared to all other species pooled (2.25%, *p* = 0.008; FET). In the sample of male *B. bimaculatus*, 30.8% were positive for neogregarines, and this was significantly higher than for workers (9.4%, *p* < 0.0001; FET). There were no differences between *B. impatiens* and *B. griseocollis* for any comparison among the parasites (Table 1 and Table 2). *Bombus vagans* and *B. perplexus* harbored 15% and 6.9% intense *Nosema* infections, respectively, but high *Nosema* intensity was not observed in the three other species (Table 2). Intense trypanosomatid infections were observed in all species except *B. griseocollis*, ranging from 1.9% intense infections in *B. impatiens* to 9.1% and 10.5% in *B. perplexus* and *B. vagans,* respectively.

Although nematodes were dissected from only 13 bees (1.3%), significantly more were recorded in *B. bimaculatus* (4.7%) when compared to all other species combined (0.9%, *p* = 0.0084; FET). Conopid prevalence over the season was 14.3%. Prevalence of conopid larvae was significantly lower in the two earliest species, *B. bimaculatus* and *B. perplexus* (6.8%) as compared to the combined later-season species *B. griseocollis, B. impatiens* and *B. vagans* (16.2%, *p* = 0.0003; FET).

## 4. Discussion

The results presented here confirm that the rapidly increasing *B. impatiens* populations in our study region have considerably lower parasite burdens than species with steady or declining populations. Light parasite burden may prove to be a signal of continued regional expansion of another adaptable species; *B. griseocollis* was found to also be relatively parasite-free. Although historically absent in our study region, in past decades, *B. griseocollis* prevalence has been increasing [12], and this species is widespread elsewhere and is commonly reported in surveys of urban environments [17,47], isolated western MA meadows [29], and nearby regions such as an Atlantic coastal island [48]. With the exception of *B. bimaculatus,* where the parasite pattern differs somewhat from other species, the varying levels of parasites that we recorded are consistent with the species status derived from our long-term survey. Owing to their eusociality and frequency of monandry, when in decline, bumble bee species in isolated and fragmented populations are particularly at risk for inbreeding and loss of genetic diversity, and thus, may exhibit increased disease susceptibility [49,50]. We have lately seen a reduction in both distribution and prevalence of *B. perplexus*, and it will be instructive if the elevated pathogen load, particularly high-intensity microparasite infections, is an indicator of future decline. Of greater concern, the distribution and abundance of *B. vagans* have declined sharply in our collections, and this trend has been identified by other studies in northeast USA [2,5], and across North America (IUCN Red List http://www.iucnredlist.org, accessed on 8 June 2017). For *B. vagans,* we found both high-intensity and mixed infections, as well as elevated infection levels of *Nosema* and trypanosomatids. This result is at odds with other studies that have not detected elevated *N. bombi* infections in *B. vagans*. For example, an extensive USA-wide survey did not detect *Nosema* in any *B. vagans* [28], and in a Maine survey, where *B. vagans* was abundant, prevalence of *N. bombi* was comparable to the levels across all species [51]. Regarding *B. bimaculatus*, studies have found that the abundance of this species is highly variable among survey years [2,12,13]—a trend that may be the result of the fact that *B. bimaculatus* is an early-season species and is thus more responsive to interannual variation in spring conditions and changes in floral resources. For *B. bimaculatus*, we saw elevated trypanosomatid prevalence compared to the levels observed in both *B. impatiens* and *B. griseocollis*. Resistance to trypanosomatid infection has been correlated with the composition of gut microbiota, with diversity and abundance of bacteria being important factors [52]. For example, in New Jersey (USA), Cariveau et al. [53] found that the microbiota differed in *B. bimaculatus* compared to co-distributed *B. impatiens* and *B. griseocollis*. Moreover, for *B. bimaculatus,* the assemblage of bacteria differed according to the bee’s habitat, while this was not the case for *B. impatiens* and *B. griseocollis*. For *B. bimaculatus* in the present study, we saw significantly higher prevalence of both neogregarines and nematodes when compared to the pool of all other species. Although the nematode numbers were very low, we make no speculation regarding any aspect of their significance, but include them in this report to serve as a baseline for future parasite surveys. Surprisingly, we did not observe *Nosema* in any *B. bimaculatus* males or workers, even though individuals were collected at the same flowers as infected species. The worker sample was large, and individuals were collected across dozens of sites, suggesting that this species in this regional collection is resistant to *Nosema* [54].

All species at our study sites had medium tongue lengths (tongue length may partition *Bombus* species among available flowers) and were sharing a limited array of available flowers. Dissimilar parasite burdens (e.g., 2- to 4-fold differences in *Nosema* and trypanosomatid rates) may influence foraging performance and play a role in setting up winning and losing species that engage in asymmetric contests for pollen and nectar [1]. Individuals harboring high infection levels of trypanosomatids such as *C. bombi*, a parasite that is known to impair cognitive processes and thus impact foraging efficiency, could place colonies at a distinct disadvantage [35,36,37,38,39,40,41]. This possibility merits further study, as does likely competition among variously infected spring queens seeking to secure below-ground nests, particularly deserted rodent nests, a limiting resource [19,32,55]. In addition, future studies should utilize PCR-based approaches for obtaining species-level identifications of potential parasites, as PCR is both more sensitive for detecting parasite presence, and more accurate for species identifications [56,57,58].

Widespread acceptance that certain *Bombus* species declined as a result of exotic or native pathogen spillover from commercial *Bombus* colonies was based on circumstantial evidence from other regions [26,59]. Ready adoption of the theory was possible owing to three gaps in our baseline: (1) rarity of studies detailing comparative susceptibilities among North American *Bombus* species, (2) no historical baselines of parasite prevalence, and (3) lack of repeated measure surveys of relative and absolute abundances of *Bombus* at sites over time. Regarding the first gap, comparative susceptibilities remain largely unstudied. We are at a particular loss because the majority of research from which to draw is on European species or commercial mass-reared colonies. In this report, we addressed the second and third gaps and established a parasite baseline upon which we can overlay past shifts in community composition and to which we can refer in the future. It is unfortunate that such baselines were not carried out years earlier in order to understand the factors responsible for the drastic loss of bumble bee species in our region.

## 5. Conclusions

We investigated parasite loads in bumble bee communities where we have previously documented a regional shift in species composition. This shift has been accompanied by the gradual domination by a single species, *B. impatiens*, both in relative and absolute abundance. Much work has focused on the factors that may lead to pollinator decline, including exposure to pesticides, a lack of flowers, changing landscapes, changing climatic conditions, and especially, parasites. Less emphasis has been placed on determining why some species are thriving. We studied a key stressor, parasite load, by sampling communities throughout the season and across a region. Compared to congeners in decline, we documented a significantly lower microparasite load for the highly successful species, *B. impatiens*, and speculate that lower parasite burden in these flourishing colonies may produce stronger competitors for floral and nest resources, hastening the loss of community diversity.

## Figures and Tables

**Figure 1 insects-12-00941-f001:**
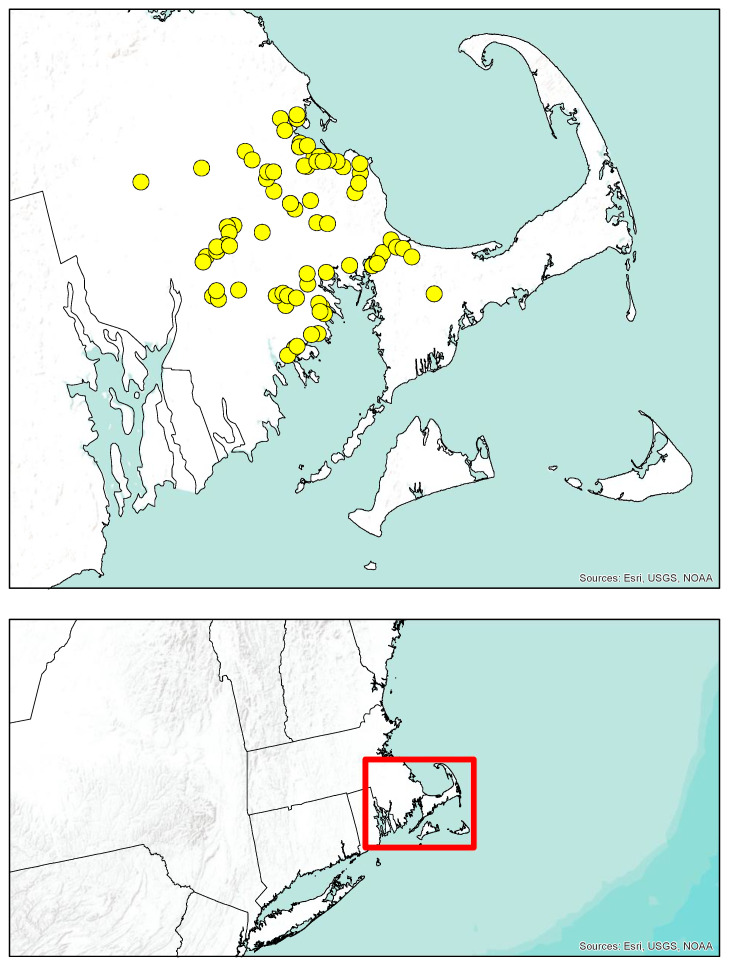
Map depicting collection locations (circles) of the 73 sites where *Bombus* workers and males were sampled across southeastern Massachusetts.

**Figure 2 insects-12-00941-f002:**
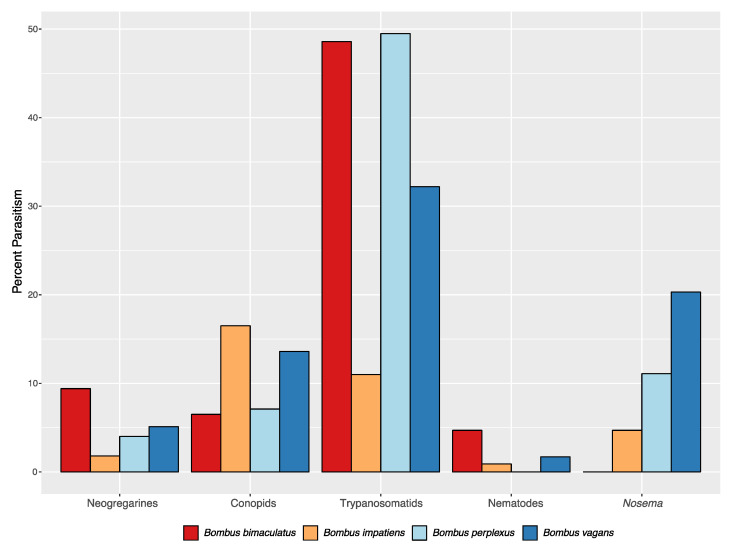
Percent infection of focal *Bombus* species workers by different parasite groups.

**Table 1 insects-12-00941-t001:** Changes in the percent of sites where species was collected (Percent Change) and relative abundance (Change in Abundance) revealed in a previously reported long-term survey (1990–2017) in our study region [11,12], and data from the present study showing the percent of sites where each species was collected (Percent of Sites), the total number of each species collected (*N*), the percent of samples with microparasites (Percent Microparasites), the percent of samples with high-intensity infections (Percent High Intensity), and the percent of samples with multiple infections (Percent Mixed).

*S*pecies	Percent Change	Change in Relative Abundance	Percent of Sites	*N*	Percent Microparasites	Percent High Intensity	Percent Mixed
*Bombus griseocollis*	+60	+3.5	10.4	20	10.5	0	0
*Bombus impatiens*	+20	+42.1	87.0	902	13.2	2.7	2.9
*Bombus bimaculatus*	+10	−9.3	46.1	120	55.2	11.2	12.6
*Bombus perplexus*	−50	−7.3	40.9	99	56.6	17.1	9.8
*Bombus vagans*	−60	−10.4	21.7	61	52.5	17.0	14.5

**Table 2 insects-12-00941-t002:** For each species, the numbers of workers and males collected (caste) during our surveys, including the percent of individuals infected with parasites. The prevalence of intense infections is shown in parentheses.

Species	Caste	*N*	Percent trypanosomatid	Percent neogregarine	Percent *Nosema*	Percent Conopid	Percent Nematode
*Pyrobombus* subgenus							
*Bombus impatiens*	Worker	710	11.0 (1.9)	1.8 (0.3)	4.7 (0)	16.5	0.9
*Bombus impatiens*	Male	192	2.1 (1.0)	0 (0)	1.1 (0)	2.6	0
*Bombus bimaculatus*	Worker	107	48.6 (9.1)	9.4 (0.9)	0 (0)	6.5	4.7
*Bombus bimaculatus*	Male	13	21.4 (15.4)	30.8 (0)	0 (0)	14.3	15.4
*Bombus perplexus*	Worker	99	49.5 (10.5)	4.0 (0)	11.1 (6.0)	7.1	0
*Bombus perplexus*	Male	0	--	--	--	--	--
*Bombus vagans*	Worker	59	32.2 (5.3)	5.1 (1.7)	20.3 (15.0)	13.6	1.7
*Bombus vagans*	Male	2	0 (0)	50.0 (0)	50.0 (50)	0	0
*Cullumanobombus* subgenus							
*Bombus griseocollis*	Worker	19	10.5 (0)	0 (0)	5.3 (0)	15.8	0
*Bombus griseocollis*	Male	1	--	--	--	--	--
*Thoracobombus* subgenus							
*Bombus fervidus*	Worker	3	0 (0)	0 (0)	66.7 (0)	0	0
*Bombus fervidus*	Male	0	--	--	--	--	--

## Data Availability

DNA sequence data are provided through GenBank (accession numbers OK044437-65).

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
