# Peer review of "Parasite Prevalence May Drive the Biotic Impoverishment of New England (USA) Bumble Bee Communities"

_insects, 2021, doi:10.3390/insects12100941_

Round 1
Reviewer 1 Report
I would like to thank the authors for this very interesting paper which deals with a hot topic in the context of pollinator decline. Obviously as the authors are english native speakers, the MS is very well written with only a few typographical errors. However, I have major issues about the methodology and statistical analysis as well as the presentation of the results. I attach here a series of remarks/suggestions.
Title : The title is a bit long. I suggest that the authors shorten it to give more impact to their study.
L60-70. The authors should also discuss other hypotheses to explain the expansion of B. impatiens. Climate change, resistance to extreme weather events. Parallelism with B. terrestris in Europe (other commercial species)?
L83-84. OK but B. terrestris is a very resistant species and some studies suggest that bumblebees could forage on plants rich in flavonoids acting as a "natural medicine" against "parasites".
L88-89. What preliminary data?
L.95-97 lacks dates of species descriptors
L97-99 authors should detail their preliminary results
Table 1. What time frame do the authors use for their comparison? Was the parasite load measured for all specimens? Pseudo-replicas? Difference between workers, queens and males. The table is not clear. Why mention affinis and terricola? Sampling bias due to phenology, collection site?
L.110 Sampling map ? Landscape photo? Foraging plants ?
L122. There is a lot of information missing regarding the collection protocol. Same collector for all sites? How many visits per site? What is the weather? How far apart are the sites? How many sites per day? If the data was acquired in 2015, did the authors not collect the other years to have temporal replicas?
L138. Identification guide to North American bumblebees (Williams et al., 2015) ?
L155-162. If you haven't measured this then this part is useless
L163-164. Why did you dissect the specimens instead of inspecting the faeces of the individuals as is usually the case
L171-174. It would have been possible not to kill the individuals to have a dynamic study by inspecting the parasite load in the faeces.
Statistical analysis non-existent !!! Which statistical test, which data processing?
L.187. Table in supplementary material with all information on collected specimens?
L201. What statistical test? No figure to visualize the results? The MS has a big gap in the illustration of the results
Did the authors do an "arcsin" transformation of the percentages before statistical processing as they should do?
L232-237. Given the low sample size for nematodes, I suggest that the authors delete this part of the experiment because with such a low sample size, the interpretations are too speculative.
L250 What hypothesis could explain the results for B. bimaculatus?
Impact of commercial strains for B. impatiens?
L262. Speculative, please elaborate on the link between genetic diversity and parasite resistance
L281-283 What are the hypotheses? Difference in diet?
L284-297. Interesting but is this really the subject of the study. I suggest deleting this part or reducing it significantly.
L297-306 OK interesting and relevant but L 306-3011 same comment as above.
L327-339. I think this part is really redundant and not useful. It would be stronger in terms of impact to end the MS on the last paragraph of the discussion possibly including 1-2 sentences of the current Conclusion. I therefore suggest deleting the conclusion part.
"Averill, A.L. (University of Massachusetts Amherst, Amherst, Massachusetts, United States of America). Unpublished data. 2021." ... impossible to verify the data
Author Response
Reviewer 1:
I would like to thank the authors for this very interesting paper which deals with a hot topic in the context of pollinator decline. Obviously as the authors are english native speakers, the MS is very well written with only a few typographical errors. However, I have major issues about the methodology and statistical analysis as well as the presentation of the results. I attach here a series of remarks/suggestions.
Response: We are extremely grateful for the reviewer’s comments and detailed edits. Below we address each point specifically.
Title : The title is a bit long. I suggest that the authors shorten it to give more impact to their study.
Response: We have shortened the title as suggested.
L60-70. The authors should also discuss other hypotheses to explain the expansion of B. impatiens. Climate change, resistance to extreme weather events. Parallelism with B. terrestris in Europe (other commercial species)?
Response: As suggested, we have added additional background information.
L83-84. OK but B. terrestris is a very resistant species and some studies suggest that bumblebees could forage on plants rich in flavonoids acting as a "natural medicine" against "parasites".
Response: As requested, we have added background information on B. terrestris. The ‘self-medication concept’, however; while interesting is outside the scope of this study.
L88-89. What preliminary data?
Response: We have clarified that this project is part of a larger long-term study of bumble bee species in our region
L.95-97 lacks dates of species descriptors
Response: Dates have been added, our apologies.
L97-99 authors should detail their preliminary results
Response: Added, as requested above.
Table 1. What time frame do the authors use for their comparison? Was the parasite load measured for all specimens? Pseudo-replicas? Difference between workers, queens and males. The table is not clear. Why mention affinis and terricola? Sampling bias due to phenology, collection site?
Response: The table legend has been updated for clarity.
L.110 Sampling map ? Landscape photo? Foraging plants ?
Response: We have added a figure that shows the sample localities.
L122. There is a lot of information missing regarding the collection protocol. Same collector for all sites? How many visits per site? What is the weather? How far apart are the sites? How many sites per day? If the data was acquired in 2015, did the authors not collect the other years to have temporal replicas?
Response: The collection methods have been clarified to address the above concerns.
L138. Identification guide to North American bumblebees (Williams et al., 2015) ?
Response: The citation to Williams et al. 2014 has been added.
L155-162. If you haven't measured this then this part is useless
Response: The section has been removed.
L163-164. Why did you dissect the specimens instead of inspecting the faeces of the individuals as is usually the case
Response: Unfortunately, it is not possible to survey the conopid fly parasitoids using the feces collection method, and the abdomen must be dissected in order to determine their presence. Moreover, in preliminary work, we found that we were more likely to detect Apicystis spores in gut dissection over fecal collection. Finally, we do not believe that fecal collections are the usual approach when assessment across the parasite taxa is the goal.
L171-174. It would have been possible not to kill the individuals to have a dynamic study by inspecting the parasite load in the faeces.
Response: As per above, dissections were required in order to obtain information about the presence of parasitoids.
Statistical analysis non-existent !!! Which statistical test, which data processing?
Response: Our apologies that the statistical methods were not more clearly presented. For all comparisons between species, we conducted Fisher’s Exact Test due to the fact that for several species the expected values were below 5 individuals (the threshold at which Chi-squared tests become more appropriate). This text is now more clearly visible in the methods, and at each point in the results where P-values from these analyses are presented, we have added the following text “FET”.
L.187. Table in supplementary material with all information on collected specimens?
Response: As requested previously, we have added a figure showing the collection locality information.
L201. What statistical test? No figure to visualize the results? The MS has a big gap in the illustration of the results
Did the authors do an "arcsin" transformation of the percentages before statistical processing as they should do?
Response: As noted above we used Fisher’s Exact Text, and have made this clearer in the text. We have also added a figure showing differences in proportions of infected individuals. However, we did not perform an arcsin transformation as transformations are not required for FET comparisons. In addition, arcsin transformations have been replaced by logistic regression analyses.
L232-237. Given the low sample size for nematodes, I suggest that the authors delete this part of the experiment because with such a low sample size, the interpretations are too speculative.
Response: We have refocused this section of the results to simply report that nematodes were collected at low numbers, and have removed the emphasis on comparisons between species. We note here that our careful assessment of nematodes could be important in future and note in the paper’s text that this provides a baseline for future study. Introduction could happen - Farmers import nematodes for pest management; moreover, imported Bombus impatiens could introduce macroparasites such as nematodes, Long shots yes, but good to record what the current levels are
L250 What hypothesis could explain the results for B. bimaculatus?
Response: Unfortunately, we have no idea, though we encourage future research into the factors that might be responsible for these differences!
Impact of commercial strains for B. impatiens?
Response: We have added information about this to the IntroductionL262. Speculative, please elaborate on the link between genetic diversity and parasite resistance
Response: We have added the following text to address this result, “Owing to their eusociality and frequency of monandry, when in decline, bumble bee species in isolated and fragmented populations are particularly at risk for inbreeding and loss of genetic diversity, and thus, may exhibit increased diseased susceptibility [47]”
L281-283 What are the hypotheses? Difference in diet?
Response: Unfortunately, we can only speculate as to what might be causing this correlation, though we encourage future research into the factors that might be responsible for these differences!
L284-297. Interesting but is this really the subject of the study. I suggest deleting this part or reducing it significantly.
Response: We removed the paragraph in question.
L297-306 OK interesting and relevant but L 306-3011 same comment as above.
Response: While we think the speculation is useful, we have reduced the focus on this in the text.
L327-339. I think this part is really redundant and not useful. It would be stronger in terms of impact to end the MS on the last paragraph of the discussion possibly including 1-2 sentences of the current Conclusion. I therefore suggest deleting the conclusion part.
Response: The Conclusion section has been re-written.
"Averill, A.L. (University of Massachusetts Amherst, Amherst, Massachusetts, United States of America). Unpublished data. 2021." ... impossible to verify the data
Response: The citation has been removed.
Reviewer 2 Report
The authors examined differences in the presence of parasites in different species of Bombus collected in coastal-zone region of New England (USA). They found contrasting parasite burden between co-occurring winners and losers in this community and conclude that this difference may impact the endgame of asymmetric contests among species competing for dwindling resources.
I think this work is interesting and deserves being published in this journal. My only major concern is on the identification of the parasites. The authors determine whether or not the bees were infected with microscopic examination. I am wondering the necessary to determine the parasites in representative samples with molecular methods. How accuracy of the microscopic examination on identification of the parasites?
A few more minor comments.
- Line 139-162. This is too much background information in Method section. I suggest incorporate these sentences into Introduction section or Discussion section.
- In table 1 and table 2, I suggest use the full name of the species.
- I think the conclusion is a little far from the research itself. I suggest rewrite this part.
Author Response
Reviewer 2:
The authors examined differences in the presence of parasites in different species of Bombus collected in coastal-zone region of New England (USA). They found contrasting parasite burden between co-occurring winners and losers in this community and conclude that this difference may impact the endgame of asymmetric contests among species competing for dwindling resources.
I think this work is interesting and deserves being published in this journal. My only major concern is on the identification of the parasites. The authors determine whether or not the bees were infected with microscopic examination. I am wondering the necessary to determine the parasites in representative samples with molecular methods. How accuracy of the microscopic examination on identification of the parasites?
Response: We are extremely grateful for the reviewer’s positive comments and suggestions. In regards to the molecular methods, we agree that that approach could likely be more accurate than the approach we utilized, and will use molecular ID’s (and the faeces approach suggested by Reviewer 1) in future studies. Below we address specific corrections.
A few more minor comments.
- Line 139-162. This is too much background information in Method section. I suggest incorporate these sentences into Introduction section or Discussion section.
Response: The text has been simplified as requested.
- In table 1 and table 2, I suggest use the full name of the species.
Response: Updated as suggested
- I think the conclusion is a little far from the research itself. I suggest rewrite this part.
Response: As suggested by Reviewer 1, this section has been rewritten.
Round 2
Reviewer 1 Report
I would like to thank the authors for their responses and for the modifications in the MS.
However, I still have some issues with the illustration of the results. Although the map is useful, I think the quality of the illustration could be much improved. Also, the authors mention that they have added a figure to show the difference in parasite load but this illustration is not included in the PDF that I was able to see, so unfortunately I cannot evaluate this point and it is very unfortunate.
Author Response
Reviewer 1: I would like to thank the authors for their responses and for the modifications in the MS. However, I still have some issues with the illustration of the results. Although the map is useful, I think the quality of the illustration could be much improved. Also, the authors mention that they have added a figure to show the difference in parasite load but this illustration is not included in the PDF that I was able to see, so unfortunately I cannot evaluate this point and it is very unfortunate.
Response: We would again like to thank the Reviewer for their thoughtful edits and comments. We have updated the text to include a redrawn version of the map to improve clarity as suggested. We have also included Figure 2 in this submission, our apologies that it was not included in the R1 submission.
Round 3
Reviewer 1 Report
I endorse the publication of this version
Author Response
Thank you for all of your constructive comments! We very much appreciate the effort you have put into improving our manuscript.